# POTENTIAL FLOW GENERATOR WITH $L_2$ OPTIMAL TRANSPORT REGULARITY FOR GENERATIVE MODELS

## ABSTRACT

We propose a potential flow generator with $L_2$ optimal transport regularity, which can be easily integrated into a wide range of generative models including different versions of GANs and normalizing flow models. With only a slight augmentation to the original generator loss functions, our generator not only tries to transport the input distribution to the target one, but also aims to find the one with minimum $L_2$ transport cost. We show the effectiveness of our method in several 2D problems, and illustrate the concept of "proximity" due to the $L_2$ optimal transport regularity. Subsequently, we demonstrate the effectiveness of the potential flow generator in image translation tasks with unpaired training data from the MNIST dataset and the CelebA dataset with a comparison against vanilla WGAN-GP and CycleGAN.

## 1 INTRODUCTION

Many of the generative models, for example, generative adversarial networks (GANs) (Goodfellow et al., 2014; Arjovsky et al., 2017; Salimans et al., 2018) and normalizing flow models (Rezende & Mohamed, 2015; Kingma & Dhariwal, 2018; Chen et al., 2018), aim to find a generator that could map the input distribution to the target distribution.

In many cases, especially when the input distributions are purely noises, the specific maps between input and output are of little importance as long as the generated distributions match the target ones. However, in other cases like image-to-image translations, where both input and target distributions are distributions of images, the generators are required to have additional regularity such that the input individuals are mapped to the "corresponding" outputs in some sense. If paired input-output samples are provided, $L_p$ penalty could be hybridized into generators loss functions to encourage the output individuals to fit the ground truth (Isola et al., 2017). For the cases without paired data, a popular approach is to introduce another generator and encourage the two generators to be the inverse maps of each other, as in CycleGAN (Zhu et al., 2017), DualGAN (Yi et al., 2017) and DiscoGAN (Kim et al., 2017), etc. However, such a pair of generators is not unique and lacks clear mathematical interpretation about its effectiveness.

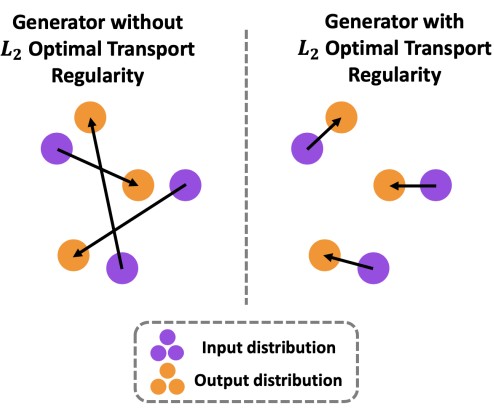

Figure 1: Schematic of generator without and with $L_2$ optimal transport regularity.

In this paper we introduce a special generator, i.e., the potential flow generator, with $L_2$ optimal transport regularity. By applying such generator, not only are we trying to find a map from the input distribution to the target one, but we also aim to find the *optimal transport map* that minimizes the squared Euclidean transport distance. In Figure 1 we provide a schematic comparison between generators with and without optimal transport regularity. While both generators provide a scheme to map from the input distribution to the output distribution, the total squared transport distances in the

left generator is larger than that in the right generator. Note that the generator with optimal transport regularity has the characteristic of "proximity" in that the inputs tend to be mapped to nearby outputs. As we will show later, this "proximity" characteristic of $L_2$ optimal transport regularity could be utilized in image translation tasks. Compared with other approaches like CycleGAN, the $L_2$ optimal transport regularity has a much clearer mathematical interpretation.

There have been other approaches to learn the optimal transport map in generative models. For example, Seguy et al. (2017) proposed to first learn the regularized optimal transport plan and then the optimal transport map, based on the dual form of regularized optimal transport problem. Also, Yang & Uhler (2018) proposed to learn the unbalanced optimal transport plan in an adversarial way derived from a convex conjugate representation of divergences. In the W2GAN model proposed by Leygonie et al. (2019), the discriminator's objective is the 2-Wasserstein metric so that the generator is supposed to recover the $L_2$ optimal transport map. All the above approaches need to introduce, and are limited to, specific loss functions to train the generators.

Our proposed potential flow generator takes a different approach in that with only a slight augmentation to the original generator loss functions, our generator could be integrated into a wide range of generative models with various generator loss functions, including different versions of GANs and normalizing flow models. This simple modification makes our method easy to adopt on various tasks considering the existing rich literature and the future developments of generative models.

In Section 2 we present a formal definition of optimal transport map and the motivation to apply $L_2$ optimal transport regularity to generators. In Section 3 we give a detailed formulation of the potential flow generator and the augmentation to the original loss functions. Results are then provided in Section 4. We include the discussion and conclusions in Section 5.

## 2 GENERATIVE MODELS AND OPTIMAL TRANSPORT MAP

First, we introduce the concept of *push forward*, which will be used extensively in the paper.

**Definition 1** *Given two Polish space $\mathbb{X}$ and $\mathbb{Y}$, $\mathcal{B}(\mathbb{X})$ and $\mathcal{B}(\mathbb{Y})$ the Borel $\sigma$-algebra on $\mathbb{X}$ and $\mathbb{Y}$, and $\mathcal{P}(\mathbb{X}), \mathcal{P}(\mathbb{Y})$ the set of probability measures on $\mathcal{B}(\mathbb{X})$ and $\mathcal{B}(\mathbb{Y})$. Let $f : \mathbb{X} \to \mathbb{Y}$ be a Borel map, and $\mu \in \mathcal{P}(\mathbb{X})$. We define $f_{\#}\mu \in \mathcal{P}(\mathbb{Y})$, the push forward of $\mu$ through $f$, by*

$$f_{\#}\mu(\mathbb{A}) = \mu(f^{-1}(\mathbb{A})), \forall \mathbb{A} \in \mathcal{B}(\mathbb{Y}). \tag{1}$$

With the concept of push forward, we can formulate the goal of GANs and normalizing flow models as to train the generator $G$ such that $G_{\#}\mu$ is equal to or at least close to $\nu$ in some sense, where $\mu$ and $\nu$ are the input and target distribution, respectively. Usually, the loss functions for training the generators are metrics of closeness that vary for different models. For example, in continuous normalizing flows (Chen et al., 2018), such metric of closeness is $D_{\mathrm{KL}}(G_{\#}\mu||\nu)$ or $D_{\mathrm{KL}}(\nu||G_{\#}\mu)$. In Wasserstein GANs (WGANs) (Arjovsky et al., 2017), the metric of closeness is the Wasserstein-1 distance between $G_{\#}\mu$ and $\nu$, which is estimated in a variational form with the discriminator neural network. As a result, the generator and discriminator neural networks are trained in an adversarial way:

$$\min_{G} \max_{D \text{ is 1-Lipschitz}} \mathbb{E}_{\boldsymbol{x} \sim \nu} D(\boldsymbol{x}) - \mathbb{E}_{\boldsymbol{z} \sim \mu} D(G(\boldsymbol{z})), \tag{2}$$

where $D$ is the discriminator neural network and the Lipschitz constraint could be imposed via the gradient penalty (Gulrajani et al., 2017), spectral normalization (Miyato et al., 2018), etc.

Now we introduce the concept of optimal transport map as follows:

**Definition 2** *Given a cost function $c : \mathbb{X} \times \mathbb{Y} \to \mathbb{R}$, and $\mu \in \mathcal{P}(\mathbb{X})$, $\nu \in \mathcal{P}(\mathbb{Y})$, we let $\mathbb{T}$ be the set of all transport maps from $\mu$ to $\nu$, i.e. $\mathbb{T} := \{f : f_{\#}\mu = \nu\}$. Monge's optimal transport problem is to minimize the cost functional $C(f)$ among $\mathbb{T}$, where*

$$C(f) = \mathbb{E}_{\boldsymbol{x} \sim \mu} c(\boldsymbol{x}, f(\boldsymbol{x})) \tag{3}$$

*and the minimizer $f^* \in \mathbb{T}$ is called the optimal transport map.*

In this paper, we are concerned mostly with the case where $\mathbb{X} = \mathbb{Y} = \mathbb{R}^d$ with $L_2$ transport cost, i.e., the transport $c(\boldsymbol{x}, \boldsymbol{y}) = \|\boldsymbol{x} - \boldsymbol{y}\|^2$. We assume that $\mu$ and $\nu$ are absolute continuous w.r.t. Lebesgue

measure, i.e. they have probability density functions. In general, Monge's problem could be ill-posed in that $\mathbb{T}$ could be empty set or there is no minimizer in $\mathbb{T}$. Also, the optimal transport map could be non-unique. However, for the special case we consider, there exists a unique solution to Monge's problem (Brenier, 1991; Gangbo & McCann, 1996).

Informally speaking, with $L_2$ transport cost the optimal transport map has the characteristic of "proximity", i.e. the inputs tend to be mapped to nearby outputs. In image translation tasks, such "proximity" characteristic would be helpful if we could properly embed the images into Euclidean space such that our preferred input-output pairs are close to each other. A similar idea is also proposed in Yang & Uhler (2018) for unbalanced optimal transport. Apart from image translations, the $L_2$ optimal transport problem is important in many other aspects. For example, it is closely related to gradient flow (Ambrosio et al., 2008), Fokker-Planck equations (Santambrogio, 2017), flow in porous medium (Otto, 1997), etc.

## 3 POTENTIAL FLOW GENERATOR

### 3.1 POTENTIAL FLOW FORMULATION OF OPTIMAL TRANSPORT MAP

We assume that $\mu$ and $\nu$ have probability density $\rho_\mu$ and $\rho_\nu$, respectively, and consider all smooth enough density fields $\rho(t, \boldsymbol{x})$ and velocity fields $\boldsymbol{v}(t, \boldsymbol{x})$, where $t \in [0, T]$, subject to the continuity equation as well as initial and final conditions

$$\partial_t \rho + \nabla \cdot (\rho \boldsymbol{v}) = 0,$$
$$\rho(0, \cdot) = \rho_\mu, \quad \rho(T, \cdot) = \rho_\nu. \tag{4}$$

The above equation states that such velocity field will induce a transport map: we can construct an ordinary differential equation (ODE)

$$\frac{d\boldsymbol{u}}{dt} = \boldsymbol{v}(t, \boldsymbol{u}), \tag{5}$$

and the map from the initial point to the final point gives the transport map from $\mu$ to $\nu$.

As is proposed by Benamou & Brenier (2000), for the transport cost function $c(\boldsymbol{x}, \boldsymbol{y}) = \|\boldsymbol{x} - \boldsymbol{y}\|^2$, the minimal transport cost is equal to the infimum of

$$T \int_{\mathbb{R}^d} \int_0^T \rho(t, \boldsymbol{x}) |\boldsymbol{v}(t, \boldsymbol{x})|^2 d\boldsymbol{x} dt \tag{6}$$

among all $(\rho, \boldsymbol{v})$ satisfying equation (4). The optimality condition is given by

$$\boldsymbol{v}(t, \boldsymbol{x}) = \nabla \phi(t, \boldsymbol{x}), \quad \partial_t \phi + \frac{1}{2} |\nabla \phi|^2 = 0. \tag{7}$$

In other words, the optimal velocity field is actually induced from a flow with time-dependent potential $\phi(t, \boldsymbol{x})$. The use of this formulation is well-known in optimal transport community (Trigila & Tabak, 2016; Peyré et al., 2019). In this paper we integrate this formulation in the deep generative models. Instead of solving Monge's problem and find the exact $L_2$ optimal transport map, which is unrealistic due to the limited families of neural network functions as well as the errors arising from training the neural networks, our goal is to regularize the generators in a wide range of generative models, so that the generator maps could approximate the $L_2$ optimal transport map at least in low dimensional problems. The maps would also be endowed with the characteristics of "proximity" so that we can apply them to engineering problems.

### 3.2 POTENTIAL FLOW GENERATOR

The potential $\phi(t, \boldsymbol{x})$ is the key function to estimate, since the velocity field could be obtained by taking the gradient of the potential and consequently the transport map could be obtained from Equation 5. There are two strategies to use neural networks to represent $\phi$. One can take advantage of the fact that the time-dependent potential field $\phi$ is actually uniquely determined by its initial condition from Equation 7, and use a neural network to represent the initial condition of $\phi$, i.e. $\phi(0, \boldsymbol{x})$, while approximating $\phi(t, \boldsymbol{x})$ via time discretization schemes. Alternatively, one can use a neural network to represent $\phi(t, \boldsymbol{x})$ directly and later apply the PDE regularity for $\phi(t, \boldsymbol{x})$ in Equation 7. We name the generators defined in the above two approaches as *discrete* potential flow generator and *continuous* potential flow generator, respectively, and give a detailed formulation as follows.

### 3.2.1 DISCRETE POTENTIAL FLOW GENERATOR

In the discrete potential flow generator, we use the neural network $\tilde{\phi}_0(\boldsymbol{x}) : \mathbb{R}^d \to \mathbb{R}$ to represent the initial condition of $\phi(t, \boldsymbol{x})$, i.e. $\phi(0, \boldsymbol{x})$. The potential field $\phi(t, \boldsymbol{x})$ as well as the velocity field $\boldsymbol{v}(t, \boldsymbol{x})$ could then be approximated by different time discretization schemes. As an example, here we use the first-order forward Eular scheme for the simplicity of implementation. To be specific, suppose the time discretization step is $\Delta t$ and the number of total steps is $n$ with $n\Delta t = T$, then for $i = 0, 1...n$, $\phi(i\Delta t, \boldsymbol{x})$ could be represented by $\tilde{\phi}_i(\boldsymbol{x})$, where

$$\tilde{\phi}_{i+1}(\boldsymbol{x}) = \tilde{\phi}_i(\boldsymbol{x}) - \frac{\Delta t}{2}|\nabla\tilde{\phi}_i(\boldsymbol{x})|^2, \quad \text{for } i = 0, 1, 2..., n-1. \tag{8}$$

Consequently, the velocity field $\boldsymbol{v}(i\Delta t, \boldsymbol{x})$ could be represented by $\tilde{\boldsymbol{v}}_i(\boldsymbol{x})$, where

$$\tilde{\boldsymbol{v}}_i(\boldsymbol{x}) = \nabla\tilde{\phi}_i(\boldsymbol{x}), \quad \text{for } i = 0, 1...n. \tag{9}$$

Finally, we can build the transport map from Equation 5:

$$\tilde{\boldsymbol{u}}_0(\boldsymbol{x}) = \boldsymbol{x},$$
$$\tilde{\boldsymbol{u}}_{i+1}(\boldsymbol{x}) = \tilde{\boldsymbol{u}}_i(\boldsymbol{x}) + \Delta t\tilde{\boldsymbol{v}}_i(\tilde{\boldsymbol{u}}_i(\boldsymbol{x})), \text{ for } i = 0, 1, 2...n-1, \tag{10}$$

with $G(\cdot) = \tilde{\boldsymbol{u}}_n(\cdot)$ be our transport map.

The discrete potential flow generator has built-in optimal transport regularity since the optimal condition (Equation 7) is encoded in the time discretization (Equation 8). However, such discretization also introduces nested gradients, which dramatically increases the computational cost when the number of total steps $n$ is increased. In our tests, we found that even $n = 5$ is almost intractable.

### 3.2.2 CONTINUOUS POTENTIAL FLOW GENERATOR

In the continuous potential flow generator, we use the neural network $\tilde{\phi}(t, \boldsymbol{x}) : \mathbb{R} \times \mathbb{R}^d \to \mathbb{R}$ to represent $\phi(t, \boldsymbol{x})$. Consequently, the velocity field $\boldsymbol{v}(t, \boldsymbol{x})$ can be represented by $\tilde{\boldsymbol{v}}(t, \boldsymbol{x})$, where

$$\tilde{\boldsymbol{v}}(t, \boldsymbol{x}) = \nabla\tilde{\phi}(t, \boldsymbol{x}). \tag{11}$$

With the velocity field we could estimate the transport map by solving the ODE (Equation 5) using any numerical ODE solver. As an example, we can use the first-order forward Eular scheme, i.e.

$$\tilde{\boldsymbol{u}}(0, \boldsymbol{x}) = \boldsymbol{x},$$
$$\tilde{\boldsymbol{u}}((i+1)\Delta t, \boldsymbol{x}) = \tilde{\boldsymbol{u}}(i\Delta t, \boldsymbol{x}) + \Delta t\tilde{\boldsymbol{v}}(i\Delta t, \tilde{\boldsymbol{u}}(i\Delta t, \boldsymbol{x})), \text{ for } i = 0, 1, 2...n-1, \tag{12}$$

with $G(\cdot) = \tilde{\boldsymbol{u}}(T, \cdot)$ be the transport map, where $\Delta t$ is the time discretization step and $n$ is the number of total steps with $n\Delta t = T$.

In the continuous potential flow generator, increasing the number of total steps would not introduce high order differentiations, therefore we could have large $n$, for a better precision of the ODE solver. Different from the discrete potential flow generator, the optimal condition (Equation 7) is not encoded in the continuous potential flow generator, therefore we need to penalize Equation 7 in the loss function, as we will discuss in the next subsection.

One may come up with another strategy of imposing the $L_2$ optimal transport regularity: to use a vanilla generator, which is a neural network directly mapping from inputs to outputs, and penalize the $L_2$ transport cost, i.e., the loss function is

$$L_{vanilla} = L_{original} + \alpha\mathbb{E}_{x\sim\mu}\|G(\boldsymbol{x}) - \boldsymbol{x}\|^2, \tag{13}$$

where $L_{original}$ is the original loss function for the generator, and $\alpha$ is the weight for the transport penalty. We emphasize that such strategy is much inferior to penalizing Equation 7 in the continuous potential flow generator. When training the vanilla generator with $L_2$ transport penalty, no matter how we weight the $L_2$ transport cost penalty, we always have to make a trade off between "matching the generated distribution with the target one" and "reducing the transport cost" since there is always a conflict between them, and consequently $G_{\#}\mu$ will be biased towards $\mu$. On the other hand, there is no conflict between matching the distributions and penalizing Equation 7 in the continuous potential flow generator. As a consequence, the continuous potential flow generator is robust with respect to different weights for the PDE penalty. We will show this in Section 4.

### 3.3 TRAINING THE POTENTIAL FLOW GENERATOR

While the optimal condition (Equation 7) has been considered in the above two generators, the constraints of initial and final conditions have so far been neglected. However, the constraint of initial and final conditions provides the principle to train the neural network: we need to tune the parameter in the neural network $\tilde{\phi}$ so that $G_{\#}\mu$ matches $\nu$. This could be done in the fashion of GANs or normalizing flow models.

#### 3.3.1 LOSS IN GAN MODELS

For the discrete potential flow generator, since the optimal transport regularity is already built in, the loss for training $G$ is simply the GAN loss for the generator, i.e.

$$L_{D-PFG} = L_{GAN}, \tag{14}$$

where $L_{GAN}$ actually depends on the specific version of GANs. For example, if we use WGAN-GP, then $L_{GAN} = -\mathbb{E}_{\boldsymbol{z}\sim\mu}D(G(\boldsymbol{z}))$, where $D$ is the discriminator neural network.

For the continuous potential flow generator, as mentioned above, we also need to make $\tilde{\phi}$ satisify Equation 7 for the optimal transport regularity. Inspired by the applications of neural networks in solving forward and backward problems of PDEs (Lagaris et al., 1998; Raissi et al., 2017a;b; Sirignano & Spiliopoulos, 2018), we penalize the squared residual of the PDE on the so-called "residual" points. In particular, the loss for continuous potential flow generator would be

$$L_{C-PFG} = L_{GAN} + \lambda\frac{1}{N}\sum_{i=1}^{N}[\partial_t\tilde{\phi}(t_i,\boldsymbol{x}_i) + \frac{1}{2}|\nabla\tilde{\phi}(t_i,\boldsymbol{x}_i)|^2]^2, \tag{15}$$

where $\{(t_i,\boldsymbol{x}_i)\}_{i=1}^{N}$ are the residual points for estimating the residual of the PDE (Equation 7), and $\lambda$ is the weight for the PDE penalty. In this paper we set them as the points on "trajectories" of input samples, i.e.

$$\{(t_i,\boldsymbol{x}_i)\}_{i=1}^{N} = \bigcup_{i=0}^{n}\bigcup_{\boldsymbol{x}_j\in\mathbb{B}}\{(i\Delta t, \tilde{\boldsymbol{u}}(i\Delta t,\boldsymbol{x}_j))\}, \tag{16}$$

where $\mathbb{B}$ is the set of batch samples from $\mu$. Note that the coordinates of the residual points involve $\tilde{\boldsymbol{u}}$, but this should not be taken into consideration when calculating the gradient of the loss function with respect to the generator parameters.

We point out that the residual points should cover the whole spatial-temporal domain. Theoretically, only penalizing the squared residual of the PDE on "trajectories" could lead to failure in approximating the $L_2$ optimal transport map. However, in our numerical experiments, this flawed sampling strategy still works. As an improvement, in each training iteration we can perturb the trajectory points with Gaussian noise in space and uniform noise in time as residual points, so that in principle they are sampled from the whole spatial-temporal domain.

#### 3.3.2 LOSS IN NORMALIZING FLOW MODELS

Our continuous potential flow generator could be viewed as a further development of Neural ODE (Chen et al., 2018) applied to generative models, i.e. the continuous normalizing flow. The difference mainly lies in that we set the velocity as the gradient of a time-dependent potential function, and an augmented PDE loss is required for the generator. While both density matching and maximum likelihood training could be applied, here we take the latter as an example: we assume that the density of $\mu$ and samples from $\nu$ are available, and we maximize $\mathbb{E}_{\boldsymbol{y}\sim\nu}[\log p_{G_{\#}\mu}(\boldsymbol{y})]$, where $p_{G_{\#}\mu}$ is the density of $G_{\#}\mu$. Then, the loss for the continuous potential flow generator would be:

$$L_{C-PFG} = -\mathbb{E}_{\boldsymbol{y}\sim\nu}[\log p_{G_{\#}\mu}(\boldsymbol{y})] + \lambda\frac{1}{N}\sum_{i=1}^{N}[\partial_t\tilde{\phi}(t_i,\boldsymbol{x}_i) + \frac{1}{2}|\nabla\tilde{\phi}(t_i,\boldsymbol{x}_i)|^2]^2, \tag{17}$$

where as in the GAN model, $\{(t_i,\boldsymbol{x}_i)\}_{i=1}^{N}$ are the residual points for estimating the residual of PDE (Equation 7), and $\lambda$ is the weight for the PDE penalty. $\mathbb{E}_{y\sim\nu}[\log p_{G_{\#}\mu}(\boldsymbol{y})]$ could be estimated via the approach introduced in Chen et al. (2018) and we give a detailed description in Appendix B.

The discrete potential flow cannot be trivially applied in normalizing flow models since we found that the time step size is too large to calculate the density accurately.

## 4 RESULTS

### 4.1 2D PROBLEMS

In this subsection, we apply the potential flow generators to several two-dimensional problems. We mainly study the following two problems where we know analytical solutions for the optimal transport maps. In problem 1 we assume that both $\mu$ and $\nu$ are Gaussian distributions with $\mu = \mathcal{N}([0;0],[0.25,0;0,1])$ and $\nu = \mathcal{N}([0;0],[1,0;0,0.25])$. In this case the optimal transport map is $f((x,y)) = (2x,0.5y)$. In problem 2 we assume that $\mu$ and $\nu$ are concentrated on concentric rings. In polar coordinates, suppose $\mu$ has $(r,\theta)$ uniformly distributed on $[0.5,1] \times [0,2\pi)$, while $\nu$ has $(r,\theta)$ uniformly distributed on $[2,2.5] \times [0,2\pi)$, where $r$ and $\theta$ are radius and angular, respectively. In this case the optimal transport map is $f((r,\theta)) = (r+1.5,\theta)$ in polar coordinates. We present the proofs in Appendix A. Samples from $\mu$ and $\nu$ as well as the optimal transport map in both problems are illustrated in Figure 2. We prepared 40000 samples for the input distribution and target distribution in each problem as training data for Section 4.1.1 and 4.1.2, and 1000 samples for Section 4.1.3.

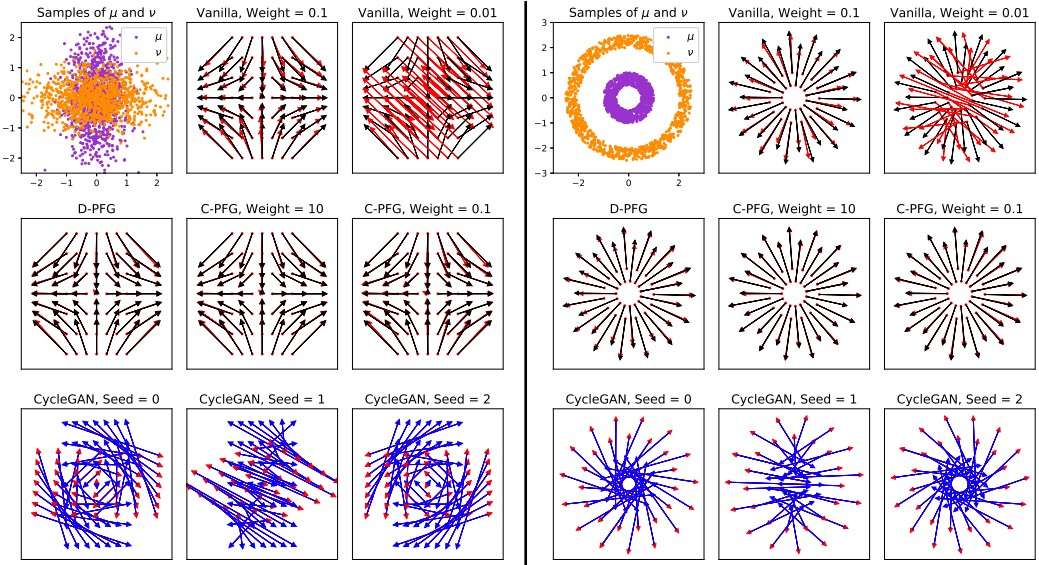

Figure 2: Comparison of different methods: vanilla generator with $L_2$ transport cost penalty, discrete potential flow generator (D-PFG), continuous potential flow generator (C-PFG), and CycleGAN in problem 1 (left) and 2 (right). The red and blue arrows represent the map of generators, the black arrows represent the analytical optimal transport map.

### 4.1.1 VANILLA GENERATOR VERSUS POTENTIAL FLOW GENERATOR

For the above two problems we compare the following methods: (a) vanilla generator with $L_2$ transport cost penalty, i.e., using the loss function in Equation 13 with GAN loss as $L_{original}$, (b) discrete potential flow generator, and (c) continuous potential flow generator with PDE penalty. For the vanilla generator and the continuous potential flow generator, we test different weights for the penalty in order to compare the influence of penalty weights in both generators. As for the GAN loss for generators we use the sliced Wasserstein distance[1], due to its relatively low computational cost, robustness, and clear mathematical interpretation in low dimensional problems (Deshpande et al., 2018). In Figure 2 we illustrate the maps of different generators. A more systematic and quantitative comparison from three independent runs for each case is provided in Table 1, where the best results are marked as bold. The statistics come from 100,000 testing data.

---

[1]Strictly speaking, there is no "adversarial" training when we use sliced Wasserstein loss since the distance is estimated explicitly rather than represented by an other neural network. However, the idea of computing the distance between fake data and real data coincides with other GANs, especially WGANs. Therefore, in this paper we view the sliced Wasserstein distance as a special version of GAN loss.

Table 1: Comparison between different generators on two problems

|  | Problem 1 | | | Problem 2 | |
|  | Std in x-axis | Std in y-axis | Error of map | Mean of norm | Error of map |
| --- | --- | --- | --- | --- | --- |
| Reference | 1.000 | 0.500 | | 2.250 | |
| Vanilla ($\alpha$=0.1) | 0.919±0.004 | 0.592±0.003 | 0.108±0.002 | 2.107±0.002 | 0.146±0.003 |
| Vanilla ($\alpha$=0.01) | 0.985±0.005 | **0.499±0.006** | 0.439±0.587 | 2.227±0.004 | 0.973±1.319 |
| Vanilla ($\alpha$=0.001) | 0.992±0.009 | 0.493±0.001 | 0.462±0.593 | 2.243±0.002 | 1.000±1.311 |
| D-PFG | **0.993±0.001** | 0.498±0.002 | **0.018±0.006** | Fail sometimes | |
| C-PFG ($\lambda$=10.0) | 0.991±0.001 | 0.502±0.001 | **0.018±0.006** | 2.243±0.001 | **0.024±0.004** |
| C-PFG ($\lambda$=1.0) | 0.992±0.001 | **0.499±0.002** | 0.019±0.007 | **2.245±0.000** | 0.029±0.002 |
| C-PFG ($\lambda$=0.1) | 0.990±0.002 | 0.503±0.003 | 0.025±0.008 | **2.245±0.001** | 0.031±0.004 |

As we already mentioned, vanilla generators with $L_2$ transport penalty would make $G_{\#}\mu$ biased towards $\mu$ to reduce the transport cost from $\mu$ to $G_{\#}\mu$. This is clearly shown in both problems with the penalty weight $\alpha = 0.1$. In fact, we observed more significant biases with larger penalty weights. For the cases with smaller penalty weights $\alpha = 0.01, 0.001$, in some of the runs, while $G_{\#}\mu$ are close to $\nu$, the maps of generators are far from the optimal ones, which shows that the $L_2$ transport penalty cannot provide sufficient regularity if the penalty weight is too small. These numerical results are consistent with our earlier discussion about the intrinsic limitation of the $L_2$ transport penalty.

On the other hand, the potential flow generators give better matching between $G_{\#}\mu$ and $\nu$, as well as smaller errors between the estimated transport maps and the analytical optimal transport maps. Notably, in both problems the continuous potential flow generators give good results with a wide range of PDE penalty weights ranging from 0.1 to 10, which shows the superiority of PDE penalty in the continuous potential flow generators compared with the transport penalty in vanilla generators. We also report that while in the first problem the discrete potential flow generator achieves a comparable result with the continuous potential flow generators, in the second problem we encountered "NAN" problems during training of the discrete potential flow generator in some of the runs. This indicates that the discrete potential flow generator is not as robust as the continuous one, which could be attributed to the high order differentiations and small total time steps $n$ in the discrete potential flow generators.

### 4.1.2 CycleGAN Versus Potential Flow Generator

We also apply CycleGAN on the above two problems, with different random seeds. Here, we use feedforward networks as generators and discriminators, with WGAN-GP for the GAN loss function, and $L_1$ loss for cycle-consistency loss with weight 5. In Figure 2 we illustrate the maps of $G$ (red arrows) and $F$ (blue arrows), i.e. the two generators in CycleGAN. The red and blue arrows overlap with opposite directions, which indicates that $G$ and $F$ are approximately the inverse map of each other, as we expected from the cycle-consistency loss. However, the maps are totally different in the three runs with different random seeds, which agrees with our discussion in Section 1 that the generator pair in CycleGAN is not unique. Moreover, the generator maps are less "regular" than the maps from the potential flow generator. Specifically, we can hardly interpret the generator maps given by CycleGAN.

### 4.1.3 Discrete Regularized Optimal Transport Solver Versus Potential Flow Generator

Finally, we compare the continuous potential flow generator in SWG and WGAN-GP with the discrete regularized optimal transport solver[2] introduced by Seguy et al. (2017) on the above two problems. The results of the output distributions, as well as the errors between estimated transport maps and the

---

[2]We used the code from https://github.com/vivienseguy/Large-Scale-OT, with the author's permission. All the setups are kept as default, except the training data and batch size. Here we use the same training dataset of size 1000 for all the methods, which is of the same magnitude as their original training dataset in the test code. Their default batch size is 50. Note that their solver is actually looking for the *regularized* optimal transport, but when the weight for regularization is small, e.g. 0.02 in their code, we expect the results to be close to the exact optimal transport.

Table 2: Comparison between discrete regularized OT solver and C-PFG on two problems

|  | Problem 1 | | | Problem 2 | |
|---|---|---|---|---|---|
|  | Std in x-axis | Std in y-axis | Error of map | Mean of norm | Error of map |
| Reference | 1.000 | 0.500 | | 2.250 | |
| RegOT, bs=50 | 0.908 | 0.444 | 0.105 | 2.185 | 0.169 |
| C-PFG, bs=50, SWG | 0.921 | 0.600 | 0.117 | 2.119 | **0.158** |
| C-PFG, bs=50, WGAN | **1.017** | **0.519** | **0.102** | **2.201** | 0.257 |
| RegOT, bs=1000 | Fail | | | Fail | |
| C-PFG, bs=1000, SWG | **1.019** | 0.509 | **0.079** | **2.237** | **0.116** |
| C-PFG, bs=1000, WGAN | 1.022 | **0.502** | 0.097 | 2.239 | 0.129 |

analytical optimal transport maps are shown in Table 2, where the best results for different batch size setups are marked as bold. The statistics come from 100,000 testing data.

We first set the batch size as 50; in this case, the errors of maps are similar for all three methods, while the output distributions of the continuous potential flow generator in WGAN-GP match the best with the target ones. As is well known, the gradients of the sample Wasserstein loss are biased (Bellemare et al., 2017), thus the output distributions are biased for GANs based on the Wasserstein loss. This problem could be serious when the batch size is small. Therefore, we increased the batch size to 1000. In this case, we encounterred "NAN" problems when learning the barycentric mapping in the discrete regularized OT solver. SWG and WGAN-GP with continuous potential flow generators are stable, and we can see an improvement in the output distributions and error of maps, as we expected.

### 4.1.4 MORE PROBLEMS IN GAN MODEL AND NORMALIZING FLOW MODEL

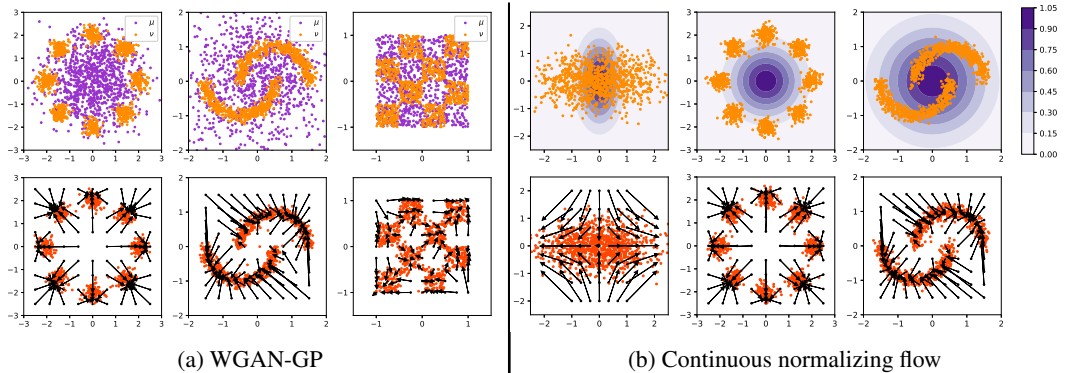

(a) WGAN-GP                           (b) Continuous normalizing flow

Figure 3: Potential flow generator in (a) WGAN-GP and (b) continuous normalizing flow for different problems. Each column shows the setup and results of one problem. The top row shows the samples or the unnormalized density functions of $\mu$ (purple) and $\nu$ (orange), the bottom row shows the map estimated by potential flow generator $G$ and samples of $G_{\#}\mu$.

Apart from the previous two problems, we also apply WGAN-GP and continuous normalizing flow models with continuous potential flow generators to several more complicated distributions. The results are illustrated in Figure 3. We can see the match between $G_{\#}\mu$ and $\nu$ in each of the problems, as well as that the samples of $\mu$ tend to be mapped to nearby positions. This shows the effectiveness of the continuous potential flow generator in various generative models, as well as the characteristics of "proximity" in the potential flow generator maps due to the $L_2$ optimal transport regularity.

### 4.2 IMAGE TRANSLATION TASKS

In this section, we aim to show the capability of continuous potential flow generator in dealing with high dimensional problems, and also to show its advantage in image translations tasks with unpaired training data. We use WGAN-GP for the GAN loss. Before feeding the images into the generators,

we embed them into a Euclidean space, where the $L_2$ distances between embedding vectors should quantify the similarities between images. In this paper we apply the principal component analysis (PCA) (Jolliffe, 2011), a simple but highly interpretable approach to conduct the image embedding.

### 4.2.1 THE MNIST DATASET

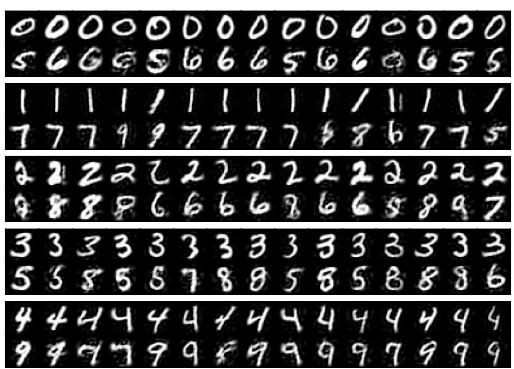

Figure 4: Potential flow generator on the MNIST. In each raw, the top images are the inputs, while the bottom images are the corresponding outputs.

We firstly test the problem of translation between the MNIST images (LeCun et al., 2010). We divide the MNIST training dataset into two clusters: (a) images of digits 0 to 4, and (b) images of digits 5 to 9. We view the two clusters of images as samples of $\mu$ and $\nu$, respectively, i.e., we want to find the optimal transport map from images of digits 0 to 4 to images of digits 5 to 9. The dimensionality of Euclidean space, i.e. the total components in PCA, is set as 100.

In Figure 4 we randomly pick images from the test dataset and show the corresponding inputs and outputs (more images in Appendix D). As we can see, the potential flow generator tends to translate images of digit 0 to digit 6, digit 1 to digit 7, digit 3 to digit 5 or 8, and digit 4 to digit 9. This is consistent with our previous discussion about the characteristics of "proximity" in that the input digits and output digits "look similar", and the corresponding embedding vectors should be close in the $L_2$ distance.

### 4.2.2 THE CELEBA DATASET

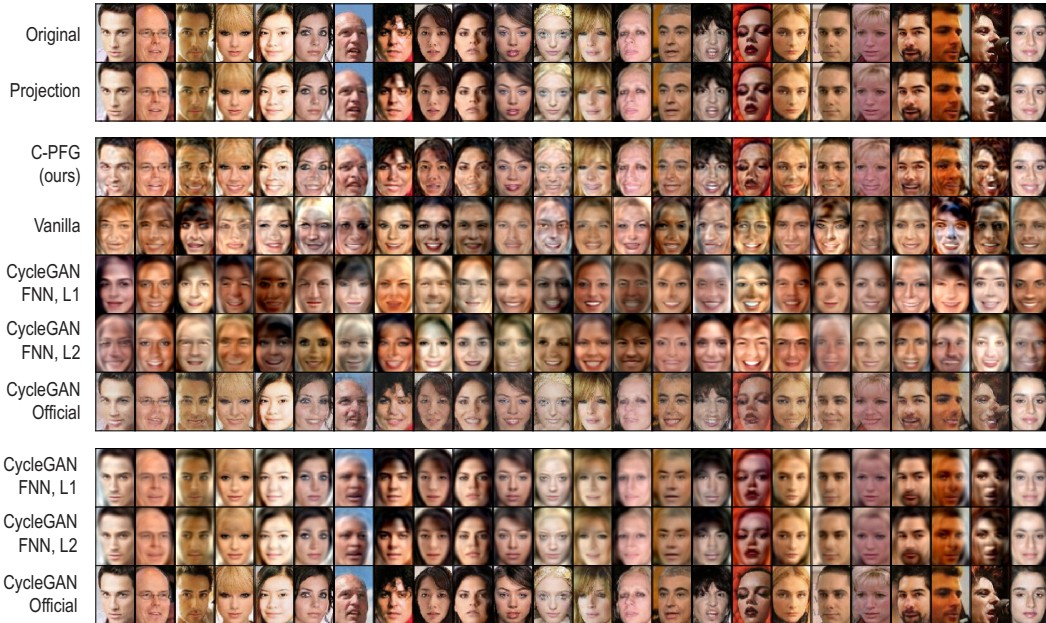

Figure 5: Comparison between our method, vanilla WGAN-GP, CycleGAN with WGAN-GP as GAN loss and $L_1$ or $L_2$ loss as cycle-consistency loss, as well as official CycleGAN. The first two rows are the original images and their projections on the 700-dimensional Euclidean space induced by PCA, which are similar. The next five rows are the *corresponding* outputs. The last three rows are the reconstructed images from different CycleGANs.

In this section we test the translation task between the CelebA images (Liu et al., 2015). We randomly pick 60000 images from the CelebA training dataset and divide them into two clusters: (a) images with attribute "smiling" labeled as false, and (b) images with attribute "smiling" labeled as true. The images are cropped so that only faces remain on the images. We view the two clusters as samples of $\mu$ and $\nu$, respectively. The total number of components in PCA is set to 700. Note that our goal is to generate images of smiling faces belonging to the same people as in input images. The difficulty lies in that the training data are unpaired and actually we do not have the ground truth as a reference.

We compare our method against vanilla WGAN-GP, and CycleGAN with WGAN-GP for GAN loss. In CycleGAN we use $L_1$ or $L_2$ loss as cycle-consistency loss, with weight = 10 as in Zhu et al. (2017). To make a fair comparison, for each generator and discriminator, we use a feedforward neural network (FNN) with the same size of hidden layers ($5 \times 256$). We also test the official CycleGAN[3] on this problem, where we feed the cropped face images (with resizing but without random jitter cropping) instead of embedding vectors into the model. The results of images randomly picked from the test dataset are shown in Figure 5.

For most of the images, our method could successfully translate the no-smiling faces to smiling faces belonging to the same people. Some of the output images are blurred, since it is difficult to learn the high order modes of PCA with FNN. Vanilla WGAN-GP and CycleGAN with FNN totally failed, in that the input and output images come from different people. This comparison clearly showed the necessity of additional regularity for the generators in translation tasks with only unpaired data, and that GAN loss + cycle-consistency loss cannot provide sufficient regularity.

The official CycleGAN is decent in performance, generating images less blurred than our method, but failed to change the countenance for some images. Note that the number of parameters in official CycleGAN is about 30 times more than that in our method, and the total training time is more than two times of our method on a single NVIDIA Tesla V100 GPU.

## 5 DISCUSSION AND CONCLUSIONS

In this paper we propose potential flow generators with $L_2$ optimal transport regularity as plug-and-play generator modules that could be easiy integrated in a wide range of generative models. In particular, we propose two versions: the discrete one and the continuous one. For the discrete version, the $L_2$ optimal transport regularity is directly encoded in, while for the continuous version we only need a slight augmentation to the original generator loss functions to impose the $L_2$ optimal transport regularity.

We firstly show that the potential flow generators are able to approximate the $L_2$ optimal transport maps in 2D problems. The continuous potential flow generator outperforms the discrete one in robustness. The continuous potential flow generator is also applied to WGAN-GP and continuous normalizing flow models, where we illustrate the characteristic of "proximity" for the potential flow generator due to the $L_2$ optimal transport regularity. Consequently, we show the effectiveness of our method in image translation tasks using unpaired training data from the MNIST dataset and the CelebA dataset. We can see that our method significantly outperforms the vanilla WGAN-GP and CycleGAN using FNN with the same size of hidden layers.

We think that the results of our method in translation tasks are impressive considering that we only use PCA, a linear embedding method, with only feedforward neural networks. Such a naive strategy actually leads to the blurred patterns in the output images, which is also the case (even more severe) for vanilla WGAN-GP and CycleGAN using the same strategy. A possible improvement is to integrate the potential flow generator with other embedding techniques like autoencoders with convolutional neural networks. Apart from image-to-image translations, it is also possible to apply the potential flow generator to other translation tasks, if the translation objects could be properly embedded into Euclidean space. Moreover, it was perceived that the training of ODE-based model is slow, but the training of our method could be accelerated by applying methods related to optimal transport, e.g. the Wasserstein natural gradient method (Tong Lin et al., 2018; Li & Montúfar, 2018). We leave these possible improvements to future work.

---

[3]We used the code from https://www.tensorflow.org/tutorials/generative/cyclegan. The size of training dataset is reduced to 1000 for official CycleGAN, which is similar to the horse-to-zebra dataset. The number of total epochs is set back to 200 as in the original CycleGAN paper.

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

## A    PROOF OF THE OPTIMAL MAPS IN SECTION 4.1

In problem 1, for Gaussian distributions $\mu$ and $\nu$ with mean $\boldsymbol{m}_1$ and $\boldsymbol{m}_2$, as well as covariance matrices $\boldsymbol{\Sigma}_1$ and $\boldsymbol{\Sigma}_2$, from Gelbrich (1990) we know that the minimum transport cost from $\mu$ to $\nu$ with cost function $c(\boldsymbol{x}, \boldsymbol{y}) = \|\boldsymbol{x} - \boldsymbol{y}\|^2$ is

$$\|\boldsymbol{m}_1 - \boldsymbol{m}_2\|^2 + \text{Tr}(\boldsymbol{\Sigma}_1 + \boldsymbol{\Sigma}_2 - 2(\boldsymbol{\Sigma}_1^{1/2}\boldsymbol{\Sigma}_2\boldsymbol{\Sigma}_1^{1/2})^{1/2}), \tag{18}$$

which is known as the squared Wasserstein-2 distance between two Gaussian distributions. In particular, the minimum transport cost is 0.5 in our problem.

For the map $f((x, y)) = (2x, 0.5y)$, $f_{\#}\mu$ is Gaussian since $f$ is linear. By checking the mean and covariance we have $f_{\#}\mu = \nu$. Also, the transport map of $f$ is

$$\mathbb{E}_{x \sim \mathcal{N}(0, 0.25)}(2x - x)^2 + \mathbb{E}_{y \sim \mathcal{N}(0, 1)}(0.5y - y)^2 = 0.25 + 0.25 = 0.5 \tag{19}$$

which is exactly the minimum transport cost, thus $f$ is the optimal transport map. We complete the proof of the optimal transport map in problem 1.

In problem 2, denote $\mathbb{X} = [0.5, 1]$, $\mathbb{Y} = [2, 2.5]$, $\mathbb{O} = [0, 2\pi)$, and $m_1 = \mathcal{U}(\mathbb{X})$, $m_2 = \mathcal{U}(\mathbb{Y})$, $m_\theta = \mathcal{U}(\mathbb{O})$, where we use $\mathcal{U}(\mathbb{A})$ to represent the uniform probability measure on set $\mathbb{A}$.

For $f((r, \theta)) = (r + 1.5, \theta)$, $\mu = \mathcal{U}(\mathbb{X}) \times \mathcal{U}(\mathbb{O})$, $\nu = \mathcal{U}(\mathbb{Y}) \times \mathcal{U}(\mathbb{O})$, we have $f_{\#}\mu = \nu$. For any transport map from $\mu$ to $\nu$, denote as $g((r, \theta)) = (g_r(r, \theta), g_\theta(r, \theta))$ in polar coordinates, we only need to show that the transport cost of $g$ is no less than the cost of $f$.

Let $h((r, \theta)) = (g_r(r, \theta), \theta)$, then the transport cost of $g$ is no less than the cost of $h$ since the transport cost is reduced for any point $(r, \theta)$.[4]

Actually we could view $g_r(r, \theta)$ as a multivalued function in $r$ so that $g_r(r, \theta)$ induces a transport *plan* $H$ from $m_1$ to $m_2$. More formally, define measure $H : \mathcal{B}(\mathbb{X} \times \mathbb{Y}) \to \mathbb{R}$ by

$$H(\mathbb{A}) = \int_{\mathbb{X}} \int_{\mathbb{Y}} \mathbf{1}_{(x,y) \in \mathbb{A}} M(x, dy) dm_1(x) = \int_{\mathbb{A}} M(x, dy) dm_1(x), \tag{20}$$

where $M(\cdot, \cdot) : \mathbb{X} \times \mathcal{B}(\mathbb{Y}) \to \mathbb{R}$ is defined by

$$M(x, \mathbb{A}) = \int_{\mathbb{O}} \mathbf{1}_{g_r(x,\theta) \in \mathbb{A}} dm_\theta(\theta). \tag{21}$$

To see $H$ is a transport plan from $m_1$ to $m_2$, we need to check:

1. $\forall \mathbb{A} \in \mathcal{B}(\mathbb{X}), H(\mathbb{A} \times \mathbb{Y}) = m_1(\mathbb{A})$.
   This is true since

   $$H(\mathbb{A} \times \mathbb{Y}) = \int_{\mathbb{A}} \int_{\mathbb{Y}} M(x, dy) dm_1(x) = \int_{\mathbb{A}} M(x, \mathbb{Y}) dm_1(x) = m_1(\mathbb{A}), \tag{22}$$

   where we utilize that $M(x, \mathbb{Y}) = 1$.

2. $\forall \mathbb{A} \in \mathcal{B}(\mathbb{Y}), H(\mathbb{X} \times \mathbb{A}) = m_2(\mathbb{A})$.
   Note that $g_{\#}\mu = \nu$, thus $\mu(g^{-1}(\mathbb{A} \times \mathbb{O})) = \nu(A \times \mathbb{O}) = m_2(\mathbb{A})$. Also,

   $$\begin{aligned}
   \mu(g^{-1}(\mathbb{A} \times \mathbb{O})) &= \int_{\mathbb{X}} \int_{\mathbb{O}} \mathbf{1}_{g(x,\theta) \in \mathbb{A} \times \mathbb{O}} dm_\theta(\theta) dm_1(x) \\
   &= \int_{\mathbb{X}} \int_{\mathbb{O}} \mathbf{1}_{g_r(x,\theta) \in \mathbb{A}} dm_\theta(\theta) dm_1(x) \\
   &= \int_{\mathbb{X}} M(x, \mathbb{A}) dm_1(x) \\
   &= H(\mathbb{X} \times \mathbb{A}).
   \end{aligned} \tag{23}$$

   Therefore $H(\mathbb{X} \times \mathbb{A}) = m_2(\mathbb{A})$.

---

[4]It's not necessary that $h_{\#}\mu = \nu$.

We also claim that the $L_2$ transport cost of $h$ equals to that of $H$. The transport cost of $h$ and $H$ are

$$
\begin{aligned}
C(h) &= \int_{\mathbb{X}} \int_{\mathbb{O}} (g_r(x, \theta) - x)^2 dm_\theta(\theta) dm_1(x), \\
C(H) &= \int_{\mathbb{X} \times \mathbb{Y}} (y - x)^2 dH = \int_{\mathbb{X}} \int_{\mathbb{Y}} (y - x)^2 M(x, dy) dm_1(x),
\end{aligned}
\tag{24}
$$

respectively. By the definition of $M$, we have $M(x, \cdot) = g_r(x, \cdot)_{\#} m_\theta$ for any $x \in \mathbb{X}$, thus

$$
\int_{\mathbb{Y}} (y - x)^2 M(x, dy) = \int_{\mathbb{O}} (g_r(x, \theta) - x)^2 dm_\theta(\theta)
\tag{25}
$$

for any $x \in X$. Therefore $C(h) = C(H)$.

Let $F(x) = x + 1.5$ be another transport plan from $m_1$ to $m_2$, clearly the $L_2$ transport cost of $f$ equals to that of $F$. Note that the transport cost of $H$ is no less than that of $F$, since the latter one is the optimal transport plan from $m_1$ to $m_2$. This complete the proof of claim that the transport cost of $g$ is no less than that of $f$, and thus the proof of the optimal transport map in problem 2.

## B    DETAILS OF LOSS FUNCTIONS IN CONTINUOUS NORMALIZING FLOWS

To estimate $\log p_{G_{\#}\mu}(\boldsymbol{y})$, we have the ODE that connects the probability density at inputs and outputs of the generator:

$$
\frac{d}{dt} \log(p(\tilde{\boldsymbol{u}}(t, \boldsymbol{x}))) = -\nabla_{\tilde{\boldsymbol{u}}} \cdot \tilde{v}(t, \tilde{\boldsymbol{u}}(t, \boldsymbol{x})) = -\Delta_{\tilde{\boldsymbol{u}}} \tilde{\phi}(t, \tilde{\boldsymbol{u}}(t, \boldsymbol{x})),
\tag{26}
$$

for all $\boldsymbol{x}$ in the support of $\mu$, where the initial probability density $p(\tilde{\boldsymbol{u}}(0, \boldsymbol{x})) = p_\mu(\boldsymbol{x})$ is the density of $\mu$ at input $\boldsymbol{x}$, while the terminal probability density $p(\tilde{\boldsymbol{u}}(T, \boldsymbol{x})) = p_{G_{\#}\mu}(G(\boldsymbol{x}))$ is the density of $G_{\#}\mu$ at output $G(\boldsymbol{x})$.

Also, we estimate $\boldsymbol{x} = G^{-1}(\boldsymbol{y})$ by solving the ODE

$$
\frac{d\boldsymbol{w}}{dt} = -\tilde{v}(T - t, \boldsymbol{w})
\tag{27}
$$

with initial condition $\boldsymbol{w}(0)$ as $\boldsymbol{y} = G(\boldsymbol{x})$ and $\boldsymbol{w}(T)$ as the corresponding $\boldsymbol{x} = G^{-1}(\boldsymbol{y})$.

For each $\boldsymbol{y}$, we can use Equation 27 to estimate the corresponding $\boldsymbol{x} = G^{-1}(\boldsymbol{y})$, and consequently $\log(p_\mu(\boldsymbol{x}))$ since we have the density of $\mu$. Then we apply Equation 26 to estimate $\log p_{G_{\#}\mu}(\boldsymbol{y})$. By sampling $\boldsymbol{y} \sim \nu$, we can estimate $\mathbb{E}_{\boldsymbol{y} \sim \nu}[\log p_{G_{\#}\mu}(\boldsymbol{y})]$. In practice, we also need to discretize Equations 26 and 27 properly. For example, we use the first-order Euler scheme in our practice. Note that when applying maximum likelihood training, the density of $\mu$ could be unnormalized, since multiplications with $p_\mu$ would merely lead to a constant difference in the loss function.

## C    NETWORKS AND HYPERPARAMETERS

Except in the regulized discrete optimal transport solver and official CycleGAN, all the neural networks are feedforward neural networks of 5 hidden layers, each of width 128 for 2D problems, or 256 in image translation tasks. For potential flow generator, the activation function is tanh for the smoothness of $\tilde{\phi}$. The activation function in vanilla generator in Figure 2 is also tanh. All the other activation functions are leaky ReLU (Maas et al., 2013) with negative slope 0.2, except in the regulized discrete optimal transport solver and official CycleGAN.

The batch size is set as 1000 for all the cases except in Table 2 and in official CycleGAN. We use 1000 random projection directions to estimate the sliced Wasserstein distances. In WGAN-GP model the coefficient for gradient penalty is 0.1, and we do 5 discriminator updates per generator update.

In potential flow generators, the time span $T$ is 1.0. We set the number of total time steps $n = 4$ in discrete potential flow generators, while $n = 100$ in continuous potential flow generators for 2D problems and $n = 10$ in image translation tasks. The PDE penalty weight $\lambda$ for continuous potential

flow generator is set as 1.0 by default, except those in the 2D problems where we compare different generators.

We use the Adam optimizer (Kingma & Ba, 2014) for all the problems. In the 2D problems, the learning rate is set as $l = 10^{-5}$, $\beta_1 = 0.5$, $\beta_2 = 0.999$ for sliced Wasserstein distance, while set as $l = 10^{-5}$, $\beta_1 = 0.5$, $\beta_2 = 0.9$ in WGAN-GP. We train the generators for $100,000$ iterations in Figure 2 and Figure 3a, and $20,000$ iterations in Table 2. In the normalizing flow model we set $l = 10^{-4}$, $\beta_1 = 0.9$, $\beta_2 = 0.999$, and train the generator for $10,000$ iterations. In image translation tasks we set $l = 10^{-4}$, $\beta_1 = 0.5$, $\beta_2 = 0.9$, and train the generator for $100,000$ iterations for our method and CycleGANs with FNN, while $200,000$ iterations for vanilla GAN.

# D   MORE RESULTS ON THE MNSIT AND CELEBA DATASET

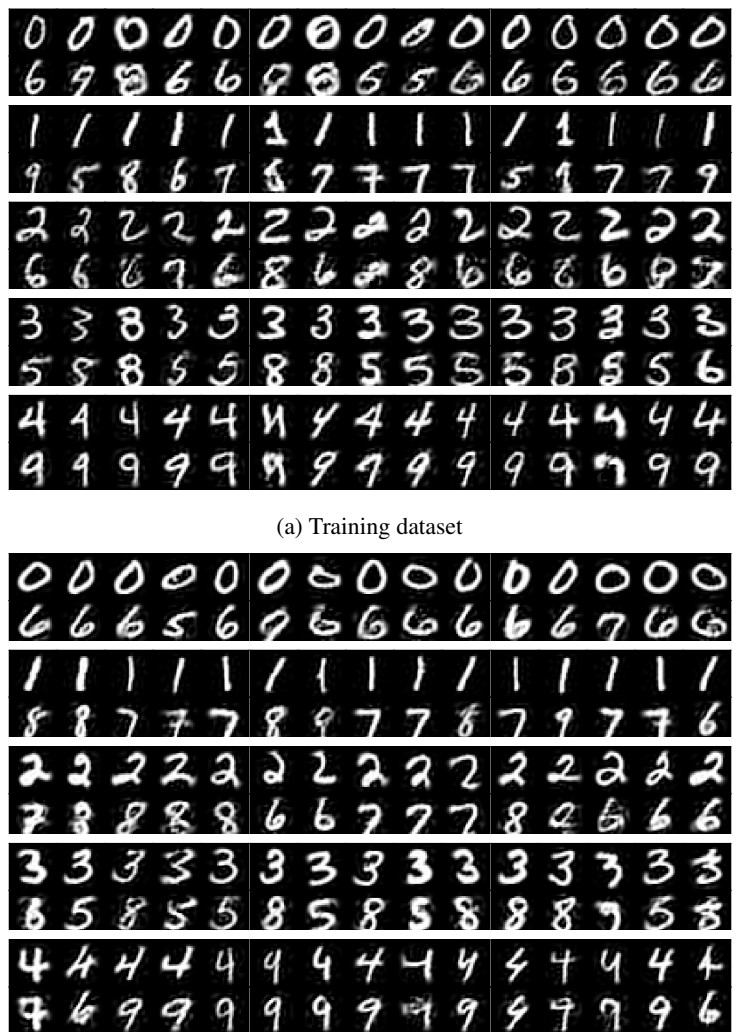

(a) Training dataset

(b) Testing dataset

Figure 6: Potential flow generator on the MNIST (a) training and (b) testing dataset. In each raw, the top images are reconstructed from the input vectors, while the bottom images are reconstructed from the corresponding output vectors.

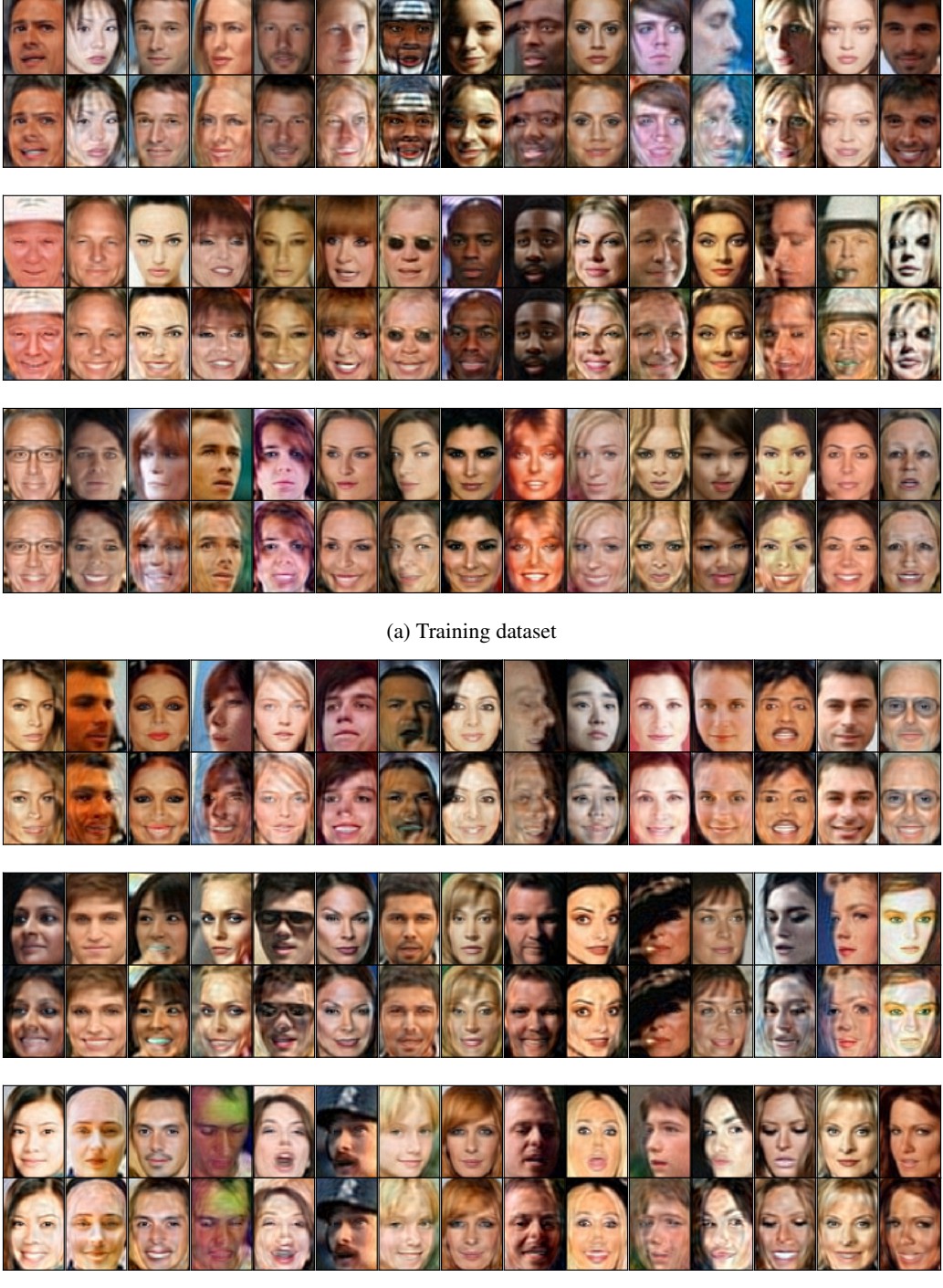

(a) Training dataset

(b) Testing dataset

Figure 7: Potential flow generator on the CelebA (a) training and (b) testing dataset. In each raw, the top images are reconstructed from the input vectors, while the bottom images are reconstructed from the corresponding output vectors.

