# OpenReview forum: "Potential Flow Generator with $L_2$ Optimal Transport Regularity for Generative Models"
_ICLR.cc/2020/Conference — Reject_

### Official Review · AnonReviewer1 · 2019-10-17
**Official Blind Review #1**

**Rating:** 3

**Review:**

The paper proposes a ‘potential flow generator’ that can be seen as a regularizer for traditional GAN losses. It is based on the idea that samples flowing from one distribution to another should follow a minimum travel cost path. This regularization is expressed as an optimal transport problem with a squared Euclidean cost. Authors rely on the dynamic formulation of OT proposed by Benamou and Brenier, 2000. They propose to learn a time-dependent potential field which gradient defines the velocity fields used to drive samples from a source distribution toward a target one. Experiments on a simple 1D case (where the optimal transport map is known), and on images with an MNIST / CelebA qualitative example.
The use of this dynamic formulation is well known in the OT community. See as a good examples:
Trigila, G., & Tabak, E. G. (2016). Data‐driven optimal transport. Communications on Pure and Applied Mathematics, 69(4), 613-648.
and more generally Chapter 7 of the ‘Computational Optimal Transport’ book by Peyré and Cuturi.

The novelty arises from the use of neural networks to represent the potentials. However, the claim that the obtained map is the optimal transport map seems wrong to me, because:
The class of potential functions over which the optimization is performed is not the whole class of functions, leading to approximations;
The optimality conditions (a.k.a continuity or preservation of mass equations) are only enforced on sampled trajectories, not on the entire space.
While this claim should definitely be lowered, it is nonetheless still acceptable provided that the proposed model is performing good. On this part, the paper strength could be improved provided that comparisons with existing methods computing a Monge map could be given. Notably, a comparison with the approach from Seguy et al. 2018 is missing. On a same level, a qualitative comparison with cycleGAN  in Figure 4 is missing.
There are some unclear elements in the paper. The final, total, optimization problem is never clearly expressed. I believe a general algorithm presentation could help in understanding the general picture of the method. Notably, for instance, It is still not clear if the generator G is disconnected from the potential definition (following Eq. 10 I assume not). How are the trajectories sampled ? Is the discriminator trained on the same sampled trajectories or different ones ?  In the potential generator, it seems that the time is considered the same way as the feature space. Can you comment on this point ? Is it possible to evaluate the flow of different time stamps that the ones used for training ?

As a summary,
Pros:
An interesting way to represent time-dependent potentials with a network for regularizing generative models
Cons:
Not much theoretical novelties in the paper, nor a good analysis on the source of errors of the model (e.g. impact of discretization on the problem)
There are some unclear aspects in the paper (see comments)
The potential benefits of the approach over the state-of-the-art should be more clearly discussed.


Minor comment:
 P3. Uniqueness of Monge problem for the squared Euclidean cost should be attributed to Brenier 91 and his polar factorization theorem. McCann generalized it to Riemannian manifold.



**Experience Assessment:**

I have published one or two papers in this area.

**Review Assessment: Checking Correctness Of Derivations And Theory:**

I assessed the sensibility of the derivations and theory.

**Review Assessment: Checking Correctness Of Experiments:**

I assessed the sensibility of the experiments.

**Review Assessment: Thoroughness In Paper Reading:**

I read the paper thoroughly.

---

> ### Author Response · Authors · 2019-11-14
> **Thanks to the reviewer for the helpful comments that helped us improve our manuscript!  (Part 1)**
>
> We would like to thank the reviewer for the helpful comments that helped us improve our manuscript!
>
> 1, The reviewer commented that the class of potential functions over which the optimization is performed is not the whole class of functions, leading to approximations;
>
> Using a neural network to approximate the potential $\phi$ will lead to errors. However, the motivation and purpose of this paper is trying to regularize the deep generative model, instead of computing the exact L2 optimal transport. We have updated the paper, especially section 3.1 and conclusion, to further clarify this point.
>
> 2, The reviewer commented that the optimality conditions are only enforced on sampled trajectories, not on the entire space.
>
> This is an excellent point, and gave us the opportunity to rethink our algorithm.
>
> Yes, the residual points should cover the whole spatial-temporal domain in principle. Theoretically, only penalizing the squared residual of the PDE on ``trajectories'' could lead to failure in approximating the L2 optimal transport map. However, in our numerical experiments, this “flawed” sampling strategy still works. As an improvement, in each training iteration we can perturb the trajectory points with Gaussian noise in space and uniform noise in time as residual points, so that they are sampled from the whole spatial-temporal domain in principle.
>
> We didn’t notice this problem in the first draft, since we thought if the solution of this HJ equation (Equation 7) exists for a certain initial condition, once it is satisfied in $supp(\rho_t)$ (in the spatial-temporal domain), then $\phi$ in $supp(\rho_t)$ should be the correct solution. However, we just realized that it’s possible that the solution doesn’t exist in the whole domain, even when the equation is satisfied in $supp(\rho_t)$.
>
> Thanks so much for pointing this out. We updated Section 3.3.1 to clarify this point.
>
> 3, About the comparisons.
>
> We have updated section 4.1.2, 4.1.3 and 4.2.2, adding the comparisons with Seguys et al.18 and CycleGAN. Thanks again for the helpful suggestions!
>
> 4, There are some unclear elements in the paper.
>
> 4.1The final, total, optimization problem is never clearly expressed. I believe a general algorithm presentation could help in understanding the general picture of the method.
> Since we are trying to design a regularized generator for a wide range of generative models, it’s difficult to give an algorithm presentation, since it depends on what kind of generative model we are using. For example, in WGAN-GP there are discriminators, but in normalizing flow model we don’t use any discriminators, etc.
> In short, our generator is a plug-and-play module in generative models: we only need to replace the original generator with ours, with a PDE loss augmented to the original generator loss (if using continuous potential flow generator). We updated the last section in the paper.
>
> 4.2 Notably, for instance, It is still not clear if the generator G is disconnected from the potential definition (following Eq. 10 I assume not) How are the trajectories sampled.
> Sorry that we didn’t make ourselves clear in the first draft.
> No, the generator is always connected from the potential definition. In short, we are using a neural network to represent the potential (see details in the reply 4.4 below). And then we have an ODE, where the velocity is the gradient of potential. Then by the time integral, we get a “trajectory” from samples of input distribution to corresponding output samples.
>
> 4.3 Is the discriminator trained on the same sampled trajectories or different ones.
> The discriminator is trained on the outputs of generators, i.e. the terminal points of sampled trajectories. We didn’t change the training of the discriminator in different generative models. Usually the training of discriminator is the key of different versions of GANs; keeping it untouched could help our method to be applied to more GAN models more easily.
>
> 4.4 In the potential generator, it seems that the time is considered the same way as the feature space. Can you comment on this point ?
> In the continuous potential flow generator, we use a neural network to represent $\phi(x,t)$, therefore the input is the concatenation of feature (spatial coordinate in Euclidean space) and time. In the discrete potential flow generator, we use a neural network to represent $\phi(x, 0)$, so the input is just the spatial coordinate. In this case $\phi(x, t)$ with t>0 should be estimated via equation 8, which is a discretization of Equation 7.
>
> 4.5 Is it possible to evaluate the flow of different time stamps that the ones used for training.
> We are not sure if we understand the question. We can evaluate the flow by sampling the “trajectories’’.

---

> > ### Author Response · Authors · 2019-11-14
> > **Part 2**
> >
> > 1, About theoretical novelty
> >
> > Yes, the mathematical formulation of optimal transport in section 3.1 is well known, but we think to integrate it in a wide range of deep generative model, and to apply it in translation tasks are new.
> >
> > 2, Error estimation
> >
> > It’s unrealistic to solve the Monge's problem and find the exact L2 optimal transport map, due to the limited families of neural network functions as well as the errors arise from training the neural networks (this might be the Holy Grail of neural networks, and is far beyond this paper). Instead, our goal is to regularize the generators in a wide range of generative models, so that the generator maps could approximate the L2 optimal transport map at least in low dimensional problems, and are endowed with the characteristics of “proximity” so that we can apply to engineering problems. We updated section 3.1 to further clarify this point.
> >
> > Minor comments:
> >
> > We thought that in Brenier’s paper in 1991, the domain is supposed to be bounded, while Gangbo and Mccann generalized to $R^d$. But you are right, Brenier should take the credit for the uniqueness of Monge problem for the squared Euclidean cost. We have updated the citation in section 2.
> >
> >
> > We thank the reviewer again for the helpful comments!

---

### Official Review · AnonReviewer2 · 2019-10-22
**Official Blind Review #2**

**Rating:** 8

**Review:**

This is a great paper using optimal transport theory for generative and implicit models. Instead of using general vector fields, the authors apply the potential vector fields in optimal transport theory to design neural networks. The mathematics is correct with convincing examples. This brings an important mathematical connection between fluid dynamics and GANs or implicit models.

I suggest the acceptance of this paper after addressing the following minor questions.

1. Would the authors provide slightly more details about the design of networks?

2. In the literature, the author may need to cite

"A. Lin, W. Li, S. Osher, G. Montufar, Wasserstein proximal of GANs, 2018."
"W. Li, G. Montufar, Natural gradient via optimal transport, 2018"

The Wasserstein natural gradient method there may improve the computational speed of the proposed models.

In all, this is an exciting paper with many potentials in future neural network designs.


**Experience Assessment:**

I have published in this field for several years.

**Review Assessment: Checking Correctness Of Derivations And Theory:**

I carefully checked the derivations and theory.

**Review Assessment: Checking Correctness Of Experiments:**

I carefully checked the experiments.

**Review Assessment: Thoroughness In Paper Reading:**

I read the paper thoroughly.

---

> ### Author Response · Authors · 2019-11-14
> **Thanks to the reviewer for the helpful suggestions that helped us improve our manuscript!**
>
> We would like to thank the reviewer for the helpful suggestions that helped us improve our manuscript!
>
> 1, The design of networks
> We updated Appendix C to provide the detailed designs of the networks and other hyperparameters.
> Specifically, in potential flow generator models, all the neural networks (including the networks representing $\phi$ and discriminator networks) are feedforward neural networks of 5 hidden layers, each of width 256 in image translation tasks, or width 128 otherwise. The input layer dimension is D in discrete potential flow generator and D+1 in continuous potential flow generator, where D is the spatial dimension, and the output layer dimension is 1 in both. The activation function is tanh so that the represented potential function is smooth.
>
> 2, We cited the two papers at the end of the revised paper. Another referee commented that the training of Neural ODE is slow, but the training of our method could be accelerated by applying methods related to optimal transport, e.g. the Wasserstein natural gradient method. We leave these possible improvements to future work.
>
> Thanks again for the helpful suggestions!

---

### Official Review · AnonReviewer3 · 2019-10-23
**Official Blind Review #3**

**Rating:** 3

**Review:**

######## Updated Review ############

The author(s) have presented a sincere rebuttal, which I really appreciate. Although I still don't quite agree with all the points made by the author(s), I have changed my mind to be more or less borderline about this submission, given that the author(s) have gone through great length to clarify and improve their manuscript.


#################################

This paper proposed a generative modeling framework called potential flow generator. Instead of deriving new matching criteria between distributions, the authors considered redefining the generative process via simulating a continuous flow that is constrained by the optimality conditions on the flow potential field derived based on L2 optimal transport. This is certainly an interesting direction to explore, however, while the points made are valid, they are not well justified. My major criticism is that too much compromise needs to be made in order to construct such a flow generator. In practical terms, it's computationally costly and sacrifices too much of the network's flexibility. My overall evaluation for this work is a straightforward/brute-force application of well-known (but less practical) results, without proposing any remedies to the real challenges that underlie. My detailed comments are listed below:

1. It is assumed that the input distribution should have the same ambient dimensionality as the target distribution, and the continuity constraints mean that the deforming to the target can take an excruciatingly slow pace, the main obstacle faced by all flow-based constructions. This point is partly evidenced by the experiment section where none of the input distributions is far from the target. It's questionable whether this framework can efficiently perform "generative modeling", in which a simple noise distribution is pushed to a more sophisticated target distribution.

2. Relations to the Neural ODE literature is not sufficiently discussed, which I believe is closest to this work. A major drawback of Neural ODE is slow computations.

3. While the author(s) have criticized an intuitive construction of L2 transport penalty in Eqn (13), their objective Eqn (17) suffers a similar issue.

4. The experiments are weak and not convincing. First, ss mentioned in earlier comments, 2D toy transport and image translation are fairly easy tasks. Second, only qualitative results are reported, and there is no baseline model to compare with. Third, without ablation study, We can hardly verify the fact the gains are actually coming from the flow part, as vanilla GANs can also perform a similar task.

Minors: Additionally, language issues can be spotted here and there. The author(s) should more carefully proofread this manuscript. And I find it confusing that Fig 3 (a) and (b) uses different examples for WGAN and CNF.

**Experience Assessment:**

I have read many papers in this area.

**Review Assessment: Checking Correctness Of Derivations And Theory:**

I assessed the sensibility of the derivations and theory.

**Review Assessment: Checking Correctness Of Experiments:**

I carefully checked the experiments.

**Review Assessment: Thoroughness In Paper Reading:**

I read the paper at least twice and used my best judgement in assessing the paper.

---

> ### Author Response · Authors · 2019-11-14
> **Thanks to the reviewer for the helpful comments that helped us improve our manuscript! (Part 1)**
>
> We would like to thank the reviewer for the helpful comments that helped us improve our manuscript.
>
> Before going into a detailed discussion,  here we want to clarify our goal again. We are not just trying to transport from input distribution to target distribution, this is already well studied in various generative models. Note that to map from one distribution to another one, there could be too many (even infinite) point-to-point map schemes. Our goal is to regularize the generators in various generative models, so that we are trying to find the optimal map, which minimizes the squared transport distance. Such regularization could be applied to translation tasks where only UNPAIRED data are available, e.g. we have images of no smiling Alice and smiling Bob, but we don’t have images of smiling Alice.
>
> Reply to the major criticisms:
>
> 1, One major criticism is about the compromise on computational cost.
>
> We made a computational cost comparison between (a) vanilla WGAN-GP, (b) CycleGAN using WGAN-GP for loss functions, and (c) WGAN-GP with the potential flow generator (our method in the paper), on the celebA translation problem. To make a fair comparison, for each generator and discriminator, we use a feedforward neural network with the same size of hidden layers (5x256) (so that the total number of parameters used in Vanilla WGAN-GP and our method are similar whereas it is double in CycleGAN), and we use the same batch size and learning rate. We test all cases on the same single NVIDIA GeForce RTX 2080Ti GPU.
>
> We found that for each iteration, vanilla WGAN-GP is the fastest, CycleGAN is about 1.5x in time,  and continuous PFG is about 2.3x in time. As for the number of iterations required for convergence, we found that our method and CycleGAN converge after 1e5 iterations. For the Vanilla GAN we observed that the output images keep changing even after 1.9e5 iterations, so we just cut off at 2e5 iteration. We made a detailed comparison in our revised paper (Figure 5), where we show that the Vanilla GAN and  CycleGANs can generate smiling faces, but from the “wrong” people.
>
> We should point out that in the celebA problem we used n=10 as the number of timesteps in flow (it’s coarse, but worked well in this problem); this is somewhat like a ResNet with about 10 blocks.  The computational cost would definitely increase if we refine the time discretization in the potential flow generator.
>
> Overall, we didn’t see unacceptable compromise on computational cost in our practice. We think it’s totally reasonable to pay some more computational cost for a correct solution, using our method. Of course we don’t want a quick but wrong answer.

---

> > ### Author Response · Authors · 2019-11-14
> > **Part 4**
> >
> > 4, The reviewer commented that the experiments are weak and not convincing.
> >
> > We updated section 4 of the paper.
> >
> > It’s easy to transport one distribution to another. However, as we mentioned above, provided with unpaired data, it’s not trivial to find the “correct” point-to-point map (e.g. we don’t want to map from Alice’s face to Bob’s face).
> >
> > We are surprised that the reviewer thinks “vanilla GANs can also perform a similar task”. Of course it cannot find the “correct” map. It will map from Alice’s no smiling face to Bob’s smiling face. We showed that in our revised paper (Figure 5).
> >
> > We need additional regularity on the generator to find the “correct” map. Actually CycleGAN is also exploring this direction: the two generators are encouraged to be the inverse of each other by the consistency loss. However, we want to point out that the consistency loss in CycleGAN cannot provide sufficient regularity: we can still construct a pair of generators to be inverse of each other, where the first generator maps from Alice’s face to Bob’s face, and the second generator maps from Bob’s face to Alice’s face. This actually happened in our experiments, as shown in our revised paper (Figures 2 and 5).
> >
> > We think the difference between Vanilla GAN, CycleGAN, and our method, is already clear via a qualitative comparison. Commonly used quantitative criteria for GANs, like Inception Score and Fréchet Inception Distance, cannot be applied to our task, since they can only evaluate the generated distribution, but cannot evaluate the point-to-point map. And we don’t have ground truth target images (e.g., we have images of no smiling Alice and smiling Bob, but we don’t have images of smiling Alice).
> >
> > Overall, we don’t agree with the reviewer that such translation problems with unpaired data are so simple that “vanilla GANs can also perform a similar task”. We made some updates on section 4.1 and 4.2, basically to compare between vanilla GAN, CycleGAN, and GANs with the potential flow generator. We hope that these updates can make our paper more convincing.
> >
> > 5. As for Figure 3, we want to point out that if we use maximum likelihood training, for any normalizing flow model, no matter discrete or continuous, the input distribution should have positive density everywhere, otherwise the log density is going to be trouble. Therefore, the third problem for WGAN, where the input distribution is a uniform distribution on a square, cannot be solved by the normalizing flow model using maximum likelihood training. Also, the 2D Gaussian to Gaussian problem is pretty interesting and standard, so we tested CNF on this problem, and found that the performance is similar to GANs. The other two problems are the same.
> >
> >
> >
> > At last we want to talk about our remedies to the real challenges. We view the interpretability of the generative models for translation tasks with unpaired data as a significant challenge. Instead of stacking tricks based on intuition, we want to formulate the problem and solve it in a mathematically interpretable way. We believe that our paper is exploring the right path.
> >
> > We apologize again for not making our points clear in the first draft. We have improved our paper based on the helpful comments. We hope our revised paper is more convincing. If there are any other questions, please let us know.

---

> > ### Author Response · Authors · 2019-11-14
> > **Part 3**
> >
> > Reply to the detailed comments:
> >
> > 1, Input distribution should have the same ambient dimensionality as the target distribution.
> >
> > If they are not in the same dimension, we cannot define L2 optimal transport. And we think such requirement for the embedding space is reasonable for translation tasks.
> >
> > The reviewer commented that “the main obstacle faced by all flow-based constructions” is that the deforming is slow. Note that in our paper, by approximating the potential $\phi$ instead of the transport scheme, we are actually looking for the generator with the shortest squared transport distance, so that the problem of slow deforming could be alleviated. In this sense, our paper is helping the development of the flow-based constructions.
> >
> > PS: We want to comment on the reviewer’s opinion that in generative model, “a simple noise distribution is pushed to a more sophisticated target distribution”. This is how the generative models were proposed years ago, but now we are far beyond that. The input distribution can be more than noise, for example, in translation tasks. We discussed this issue in the first page of the paper.
> >
> > 2,  The reviewer commented that the relations to the Neural ODE literature is not sufficiently discussed. A major drawback of Neural ODE is slow computations.
> >
> > We made an update to section 3.3.2 to discuss the relations to Neural ODE.
> >
> > Yes, Neural ODE is slow if we refine the time discretization too much. But we still like this paper since it enlightened the future development of interpretable deep learning methods. We think our paper is an example of the exploration in this direction.
> >
> > 3, The reviewer commented that while the author(s) have criticized an intuitive construction of L2 transport penalty in Eqn (13), their objective Eqn (17) suffers a similar issue.
> >
> > We don’t agree that the object Eqn 17 suffers a similar issue.
> >
> > For the intuitive construction, the L2 transport penalty and original distribution loss (GAN loss or normalizing flow loss) are conflicted, since the L2 transport penalty achieves minimum if and only if the generator is an identity map, while the original loss achieves minimum if and only if the generator transports the input distribution to the target one. As a consequence, we always need to sacrifice both for a trade off.
> >
> > Meanwhile there is no conflict between PDE loss and original loss, since we can achieve minimum for both loss when we find the optimal transport from input to target distribution (the maximum likelihood training for normalizing flow models in equation 17 is equivalent to minimizing KL(target||output)), and this is exactly what we are looking for!
> >
> > We made a detailed quantitative comparison in 4.1.1 to justify our claim.

---

> > ### Author Response · Authors · 2019-11-14
> > **Part 2**
> >
> > 2, Another major criticism is about the network’s flexibility.
> >
> > Before responding to this comment, we would like to discuss the meaning of it:  what does  flexibility mean? To map from one distribution to another one, there could be too many (even infinite) point-to-point map schemes, as is depicted in Figure 1. Actually, we have two different “flexibilities”.
> >
> > Type-1: flexibility of distribution-to-distribution map, i.e. the generator family (the set consisting of generators with different network parameters) is sufficiently rich to approximately transport the input distribution to the target one.
> > Type-2: flexibility of point-to-point map, i.e. the generator family is sufficiently rich to approximate a specific map from points to points.
> >
> > Of course type-1 flexibility is good for us, but type-2 flexibility may not be: in translation tasks with unpaired data, taking the celebA task as an example, the type-2 flexibility is actually harmful, in that if the generator is too flexible in this sense, and thus we may map Alice’s face to Bob’s face!
> >
> > Actually, this is exactly why we need to give additional “regularization” to the generator: we want to reduce the type-2 flexibility, without hurting the type-1 flexibility.
> > Cleary PFG will reduce the type-2 flexibility, in that we are looking for the L2 optimal transport maps, ruling out/discouraging the non-optimal ones.
> > As for the second part, we claim that the potential flow generator did not hurt the type-1 flexibility. We would like to show this both from theory and experiments.
> >
> > Theory: For any input and target distribution absolutely continuous w.r.t. Lebesgue measure (i.e. the distributions have density functions), the paper of Brenier has showed the existence of the time dependent potential $\phi$. In our paper we use a neural network to approximate such potential function $\phi$ directly. We cannot see any loss in the type-1 flexibility here.
> >
> > Experiments: we don’t agree with the reviewer’s comment that “none of the input distributions is far from the target”. In Figure 3 the input and output distributions are totally different. Cases like Gaussian to muti-modal Gaussian, Gaussian to arcs, are very standard and widely used as test problems. Also, while the input digits and output digits “look similar”,  we don’t think that the distribution of digits 0-4 and distribution of digits 5-9 are similar. Our experiments in the paper already showed that our generator could handle a wide spectrum of problems.
> >
> > Overall, we conclude that our method reduced the generator flexibility, but only for type-2 flexibility. This is not a bug, it's a feature!

---

### Decision · Program_Chairs · 2019-12-19

**Decision:**

Reject

**Comment:**

This paper proposes applying potential flow generators in conjunction with L2 optimal transport regularity to favor solutions that "move" input points as little as possible to output points drawn from the target distribution.  The resulting pipeline can be effective in dealing with, among other things, image-to-image translation tasks with unpaired data.  Overall, one of the appeals of this methodology is that it can be integrated within a number of existing generative modeling paradigms (e.g., GANs, etc.).

After the rebuttal and discussion period, two reviewers maintained weak reject scores while one favored strong acceptance.  With these borderline/mixed scores, this paper was discussed at the meta-review level and the final decision was to side with the majority, noting that a revision which fully addresses reviewer comments could likely be successful at a future venue.  As one important lingering issue, R1 pointed out that the optimality conditions of the proposed approach are only enforced on sampled trajectories, not actually on the entire space.  The rebuttal concedes this point, but suggests that the method still seems to work.  But as an improvement, the suggestion is made that randomly perturbed trajectories could help to mitigate this issue.  However, no experiments were conducted using this modification, which could be helpful in building confidence in the reliability of the overall methodology.

Additionally, from my perspective the empirical validation could also be improved to help solidify the contribution in a revision.  For example, the image-to-image translation experiments with CelebA were based on a linear (PCA) embedding and feedforward networks.  It would have been nice to have seen a more sophisticated setup for this purpose (as discussed in Section 5), especially for a non-theoretical paper with an ostensibly practically-relevant algorithmic proposal.  And consistent with reviewer comments, the paper definitely needs another pass to clean up a number of small grammatical mistakes.